# Assessment of Anisotropic Acoustic Properties in Additively Manufactured Materials: Experimental, Computational, and Deep Learning Approaches

**DOI:** 10.3390/s24144488

**Published:** 2024-07-11

**Authors:** Ivan Malashin, Vadim Tynchenko, Dmitry Martysyuk, Nikita Shchipakov, Nikolay Krysko, Maxim Degtyarev, Vladimir Nelyub, Andrei Gantimurov, Aleksei Borodulin, Andrey Galinovsky

**Affiliations:** 1Bauman Moscow State Technical University, Moscow 105005, Russia; 2Far Eastern Federal University, Vladivostok 690922, Russia

**Keywords:** additive technologies, selective laser melting, ultrasonic inspection, anisotropy, deep learning

## Abstract

The influence of acoustic anisotropy on ultrasonic testing reliability poses a challenge in evaluating products from additive technologies (AT). This study investigates how elasticity constants of anisotropic materials affect defect signal amplitudes in AT products. Experimental measurements on AT samples were conducted to determine elasticity constants. Using Computational Modeling and Simulation Software (CIVA), simulations explored echo signal changes across ultrasound propagation directions. The parameters $A_{13}$ (the ratio between the velocities of ultrasonic transverse waves with vertical and horizontal polarizations at a 45-degree angle to the growth direction), $A_{3}$ (the ratio for waves at a 90-degree angle), and $A_{g}$ (the modulus of the difference between $A_{13}$ and $A_{3}$) were derived from wave velocity relationships and used to characterize acoustic anisotropy. Comparative analysis revealed a strong correlation (0.97) between the proposed anisotropy coefficient $A_{g}$ and the amplitude changes. Threshold values of $A_{g}$ were introduced to classify anisotropic materials based on observed amplitude changes in defect echo signals. In addition, a method leveraging deep learning to predict $A_{g}$ based on data from other anisotropy constants through genetic algorithm (GA)-optimized neural network (NN) architectures is proposed, offering an approach that can reduce the computational costs associated with calculating such constants.

## 1. Introduction

Additive manufacturing (AM) is a modern technology for creating complex three-dimensional objects. AM is becoming increasingly common in various industries [1,2,3]. One reason is that additive manufacturing allows for the creation of parts with complex geometric shapes that may be labor-intensive or even impossible to create by traditional manufacturing methods. Materials produced by additive technologies have fundamental differences in terms of their morphology [4] compared to materials produced by traditional methods such as casting or forging. In all cases, it is important to ensure that manufactured parts meet the required quality standards.

In order to reliably detect internal flaws in products produced by additive manufacturing, non-destructive testing (NDT) methods must be used. Depending on the stage of application, inspection of additive manufacturing products can be divided into two main types: direct inspection during the manufacturing process of the product [5], and inspection after production of the product [6]. Among the main NDT methods used to inspect the AM parts are X-ray computed tomography (XCT), eddy current testing, infrared thermography, acoustic emission, and ultrasonic testing (UT). In these methods, the most promising are XCT and ultrasonic testing [6,7,8]. These inspection methods are not interchangeable; in certain cases, combinations of methods are used for more effectively flaw detection. In this article, we study the use of ultrasonic testing for the inspection of finished AM products.

Metal AM is a rapidly advancing technology poised to transform product design across the biomedical, aerospace, automotive, marine, and offshore industries. While early adopters have demonstrated significant performance gains, a comprehensive understanding of the microstructure and mechanical properties of additively manufactured metals is lacking. In order to fully harness the design potential of metal AM, especially for structural components, it is vital to address the anisotropic and heterogeneous nature of the microstructures and mechanical properties of metal AM parts. Kok et al. [9] reviewed various metal AM technologies and discussed factors contributing to anisotropy and heterogeneity, offering insights into overcoming these challenges for improved performance and reliability.

Ultrasonic automation offers precise control over particle size and shape for metal powders used in additive manufacturing. Sridharan et al. [10] observed cavity formation and droplet generation using high-speed imaging techniques. Cavitation events were found to play a crucial role in the atomization process. Experiments with liquid aluminum produced spherical particles, with particle size influenced by vibration amplitude. Their research provides insights into ways of optimizing powder production for additive manufacturing applications.

Du et al. [11] examined the anisotropic microstructure of a nickel IN718 sample fabricated using additive manufacturing (AM) and direct laser deposition (DLD). Ultrasonic scattering measurements employing longitudinal-to-transverse (L-T) mode conversion and the continuous wavelet transform (CWT) revealed increasing anisotropy towards the AM-forged interface. These results demonstrate the method’s stability in characterizing microstructural anisotropy, suggesting its suitability for additive manufacturing quality control.

Sol et al. [12] utilized pulse-echo ultrasonic testing to investigate anisotropy in $AlSi_{10}Mg$ samples produced by selective laser melting (SLM) additive manufacturing. Various ultrasonic analyses were conducted, revealing anisotropy in both transverse wave velocity and attenuation relative to the build direction. This anisotropy was symmetric around the build direction and persisted after heat treatments. These findings suggest that transverse wave velocity and frequency-dependent attenuation are sensitive tools for detecting subtle changes in additive manufacturing products.

Thevet et al. [13] investigated the anisotropic elastic properties of a $Ti_{6}Al_{4}V$ alloy under various manufacturing methods, including additive manufacturing (AM) and wrought alloy. Using dynamic pulse-echo ultrasonic techniques, the study measured sound wave velocities to derive elastic constants and Young’s module maps. Minor anisotropy was found in AM materials, while the wrought alloy exhibited higher density and ultrasonic anisotropy due to a thin phase layer along the grain boundaries. Understanding these variations is crucial for product design.

Gou et al. [14] proposed an ultrasonic peening treatment (UPT) in three directions to refine large columnar prior-β grains and secondary α grains in the cold-metal transfer additive manufacturing process of $Ti_{6}Al_{4}V$ thin-walled structures, thereby enhancing the tensile property anisotropy. Their experimental results revealed significant grain refinement post-UPT, along with minimal surface deformation. Microstructural changes and dislocations were observed, which were attributed to the mechanical effects of ultrasonic treatment within the $\alpha^{\prime }$ dissolution temperature range. Specimens treated with UPT exhibited improved properties, including higher load capacity in nano-indentation tests along with increased ultimate tensile strength and reduced anisotropic percentage in tensile tests.

Inter-layer ultrasonic impact (UI) strengthening in wire and arc additive manufacturing (WAAM) has been found to effectively reduce anisotropy in both microstructure and mechanical properties [15]. In this study, columnar microstructures were transformed into fine and uniform equiaxial ones, leading to improved stress distribution during tensile testing. Anisotropy in the tensile and yield strength decreased significantly, from 4.2% to 1.6% and from 10.1 to 2.3%, respectively. UI treatment promoted dislocation movement, leading to substructure formation and grain refinement. It also reduced local misorientations and refined the grain size while weakening the texture orientation strength.

Several studies [6,16,17] have shown the presence of anisotropy in AM materials. The properties of anisotropic materials depends on the direction. In [2,18], the authors showed that the amplitude of a signal reflected from artificial reflectors during ultrasonic testing of TC18 titanium alloy using a phased array depends on the sound beam’s direction relative to the direction of sample growth. In other works [16,19], it was found that the velocities of compression and shear waves in Inconel 718 and AlSi10Mg materials obtained by selective laser melting (SLM) changed symmetrically when changing the direction of ultrasonic wave propagation relative to the direction of samples growth. In [16], the dependence of the ultrasonic wave speed on the direction was shown to be preserved even after heat treatment. The directional dependence of acoustic properties with respect to the growth direction may be caused by the characteristics of the crystallographic texture of additive materials. Several works [20,21] have shown that grains have a predominant orientation along the growth direction when using SLM. The crystallographic texture is noted for many materials manufactured by additive methods, for example Inconel 718, $AlSi_{10}Mg$, 316L, etc. [20,22,23].

Table 1 provides an overview of articles related to defect detection using ultrasonic and other related methods along with their limitations in the context of additive technologies (AT).

A symmetrical change in acoustic properties depending on the direction is common for anisotropic single crystals [24]. The additive materials discussed in this work are polycrystalline. Anisotropy of acoustic properties in polycrystalline materials is largely determined by the presence of a crystallographic texture. Because of the influence of the growth direction on the structure, AM materials are usually considered orthotropic [25]. The propagation of an acoustic wave in a three-dimensional anisotropic medium is generally described by the following Equation (Equation 1):(1)$$C_{ijkl}\frac{\partial^{2}u_{j}}{\partial x_{k}\partial x_{l}}=\rho \frac{\partial^{2}u_{i}}{\partial t^{2}}$$
where ρ represents the material’s density and $C_{ijkl}$ denotes its elasticity modulus tensor.

As per Equation (Equation 1), the acoustic properties of the material are contingent upon its elastic properties. From the standpoint of the elastic properties, an orthotropic material behaves in a manner akin to the crystals of an orthorhombic system [16]. Crystals in an orthorhombic system possess three mutually perpendicular axes with second-order symmetry. The tensor $C_{ijkl}$ from Equation (Equation 1) exhibits symmetry in the first and second pair of indices, and its elements can be represented as a 6×6 matrix. In the case of orthotropy or the rhomboid system, the elastic moduli matrix consists solely of the nine independent constants $C_{11},C_{22},C_{33},C_{44},C_{55},C_{66},C_{12},C_{13},C_{23}$, and can be delineated as shown below.
(2)$$\left[C\right]=\left[\begin{array}{cccccc} C_{11} & C_{12} & C_{13} & 0 & 0 & 0\\ C_{12} & C_{22} & C_{23} & 0 & 0 & 0\\ C_{13} & C_{23} & C_{33} & 0 & 0 & 0\\ 0 & 0 & 0 & C_{44} & 0 & 0\\ 0 & 0 & 0 & 0 & C_{55} & 0\\ 0 & 0 & 0 & 0 & 0 & C_{66} \end{array}\right]$$

Hence, in certain instances the nature of acoustic anisotropy in additive materials can parallel the anisotropy observed in single crystals. Ultrasonic testing of anisotropic single crystals differs significantly from ultrasonic testing of isotropic materials [6,19]. The propagation of ultrasonic waves is heavily reliant on the orientation of the sound beam relative to the crystal’s axis of symmetry. When an ultrasonic wave is incident obliquely to the sample boundary, three elastic waves typically arise, each propagating at its distinct speed. Waves propagating in an anisotropic material are not purely compressional or shear. Due to the aforementioned characteristics, when ultrasonic oscillations propagate in an anisotropic material, deviations in the acoustic field’s propagation direction relative to the principal acoustic axis may ensue, along with alterations in the focusing and energy distribution of the ultrasonic beam [17].

For additive materials, a direct correlation exists between the parameters of part growth and microstructure. Factors such as scanning speed, laser beam power, and atmospheric conditions during the AM process, among others, can impact the thermal processes occurring during layer-by-layer material deposition, consequently affecting the resultant microstructure [26,27] and the anisotropy of acoustic properties. Consequently, the degree of anisotropy among samples fabricated from the same material but under different production conditions can vary significantly.

During the development of ultrasonic testing procedures, accounting for anisotropy of the material’s properties is imperative. In most instances, the amplitude of the ultrasonic signal serves as the primary informative feature for ultrasonic testing. However, when inspected from different angles relative to the symmetry axis of properties, the impact of anisotropy on the signal amplitude reflected from flaws is contingent upon numerous factors, and is rather intricate. An approach for testing anisotropic single-crystalline materials is elucidated in [26], involving the generation of ultrasonic waves normal to the surface of the test object (TO) followed by inspection along the symmetry axis of the property. While effective for single-crystalline materials, this approach often proves impractical for ultrasonic testing of additive manufacturing products due to their geometric complexity. In such scenarios, the undesirable effects of anisotropy may impede reliable ultrasonic testing. In this study, deep learning is employed to predict $A_{g}$ based on other acoustic characteristics of the material. This approach is advantageous because deep learning models can capture complex nonlinear relationships within high-dimensional data, leading to accurate and generalizable predictions.

This article delves into the challenge of determining an effective method for evaluating the influence of property anisotropy on the characteristics of an ultrasonic wave in AM product materials. An expedient method for assessing the anisotropy of acoustic properties could facilitate preliminary evaluations for conducting reliable ultrasonic testing on additive materials.

## 2. Materials and Methods

### 2.1. Experimental Study

#### 2.1.1. Sample Description

Samples were fabricated from the heat-resistant nickel alloys Inconel 718, VG159, and EP648 using the selective laser melting (SLM) method on a Concept Laser M2 Cusing device. The growth parameters for SLM samples are presented in Table 2. Additionally, Sample No. 2 underwent further processing by hot isostatic pressing (HIP).

Samples were also fabricated from Inconel 718 and AISI 321 using direct laser melting deposition (DLMD) technology. The DLMD setup utilized an IPG Photonics LS-3 fiber laser, and the process was conducted in an argon protective environment. The growth parameters for the DLMD samples are detailed in Table 3.

The samples were designed as cubes with side lengths of 25 mm. Before the experiment, the surfaces of the samples were ground to achieve a roughness level of Rz40. The cube shape was chosen to facilitate the measurement of ultrasonic wave velocities in various directions, which is essential for calculating the coefficients of the elasticity matrix ($C_{11}$ to $C_{66}$). After measuring the necessary ultrasonic speed values, chamfers were removed at a 45° angle to create additional plane-parallel surfaces. Figure 1 shows the general view of the samples before and after processing.

Flat surfaces oriented at a 45° angle relative to directions 1–3 were also necessary for measuring the velocities of ultrasonic waves in order to calculate the elasticity matrix coefficients $C_{12}$, $C_{13}$, and $C_{23}$.

#### 2.1.2. Acoustic Characteristics Measurement

To measure the acoustic properties of additive samples, A-scans were obtained using an Omniscan MX flaw detector. Figure 2 shows an example scan for Inconel 718 SLM with a transcript showing the pulse-echo numbers. Time-of-flight measurements of echo signals were performed using the contact method with a C543SM 5 MHz direct compression-wave transducer and a V155RB direct shear-wave transducer. The time of flight of the echo signal was measured using the maximum amplitude technique. The error in measuring the speeds of ultrasonic waves corresponded to about 0.5%.

Measurements of longitudinal wave attenuation coefficients were performed using an immersion contact on cubic samples prior to chamfer removal. The measurement technique corresponded to that described in [28], method 2.

#### 2.1.3. Calculation of Elasticity Matrix Coefficients $C_{ij}$

Nine independent coefficients of the elasticity matrix $C_{ij}$ for the orthotropic case were determined for the six studied samples. These coefficients were calculated after measuring the speed of ultrasonic waves using the dependencies shown below.
(3)$$\begin{array}{cc} C_{11} & =\rho \cdot V_{1/1}^{2} \end{array}$$
(4)$$\begin{array}{cc} C_{22} & =\rho \cdot V_{2/2}^{2} \end{array}$$
(5)$$\begin{array}{cc} C_{33} & =\rho \cdot V_{3/3}^{2} \end{array}$$
(6)$$\begin{array}{cc} C_{44} & =\rho \cdot V_{2/3}^{2} \end{array}$$
(7)$$\begin{array}{cc} C_{55} & =\rho \cdot V_{1/3}^{2} \end{array}$$
(8)$$\begin{array}{cc} C_{66} & =\rho \cdot V_{1/2}^{2} \end{array}$$
(9)$$\begin{array}{cc} C_{23} & =\sqrt{\left(C_{22}+C_{44}-2\rho \cdot V_{23/23}^{2}\right)\cdot \left(C_{33}+C_{44}-2\rho \cdot V_{23/23}^{2}\right)}-C_{44} \end{array}$$
(10)$$\begin{array}{cc} C_{13} & =\sqrt{\left(C_{11}+C_{55}-2\rho \cdot V_{12/12}^{2}\right)\cdot \left(C_{33}+C_{55}-2\rho \cdot V_{12/12}^{2}\right)}-C_{55} \end{array}$$

Here, $V_{i/i}$ represents the speed of a longitudinal wave in direction *i*, $V_{i/j}$ represents the speed of a transverse wave propagating in direction *i* with polarization in direction *j*, and $V_{ij/ij}$ represents the speed of a quasi-longitudinal or quasi-transverse wave propagating and polarized in the ij plane, while ρ denotes the density of the material. A detailed description of the methodology used to calculate the coefficients $C_{ij}$ is provided in [13]. Sample density measurements were not conducted in this study; therefore, the ρ values of materials provided by the manufacturer of samples manufactured under these conditions were utilized. The values of ρ are presented in Table 4.

#### 2.1.4. Porosity Analysis

X-ray tomography (X-ray CT) was conducted using a General Electric v|tome|x m300 tomograph equipped with a tube to achieve micrometer resolution tomography. The parameters of the X-ray CT performance modes are outlined in Table 5. The results were processed using specialized VGSTUDIO MAX software employing algorithms for visualizing pores and solid inclusions. Based on the sample material and voxel size, the minimum reliably detectable pore size in all samples was approximately 40 μm.

### 2.2. Computer Modelling

#### 2.2.1. Assessment of Changes in the Amplitude of the Echo Signal Reflected from the Flaw

As previously noted, the anisotropy of acoustic properties can significantly impact test results. When the difference in signal amplitudes from identical reflectors in different ultrasound propagation directions exceeds permissible error limits, concerns arise regarding the reliability of ultrasonic testing due to uneven sensitivity. This issue has been extensively documented for Austenitic materials and welded joints [17,27,29,30]. In developing inspection techniques for Austenitic materials, a preliminary assessment of testability is conducted to determine the feasibility of reliable ultrasonic testing. Testability is typically evaluated based on several criteria, including the signal-to-noise ratio and the deviation from rectilinear ultrasonic beam propagation.

This study proposes assessing the change in the amplitude of the reflected signal from an artificial reflector when inspected in different directions as an indirect indicator of the feasibility of reliable ultrasonic testing of anisotropic additive materials. In a previous study [31], it was demonstrated using Inconel 718 SLM samples that the amplitude change of the reflected signal from a side drill hole (SDH) at a depth of 13.5 mm when inspected from different angles can reach 4 dB. Experimentally determined elasticity matrix coefficients $C_{ij}$ of Inconel 718 were used as input data for modeling in the CIVA UT Simulation Tool. A quantitative comparison of the simulation results with experimental data showed that the maximum deviation between the results did not exceed 1 dB, indicating high accuracy of the amplitude change estimates obtained from CIVA simulations using these elastic constants.

Based on these findings, it can be argued that for the samples studied in [19], the crystallographic texture significantly contributes to the change in the echo signal amplitude from the SDH. The directional dependence of the amplitude change was determined by modeling using only experimentally derived coefficients of the Inconel 718 elasticity matrix. Using the CIVA modeling package and known elasticity matrix coefficients as input data, the influence of the material’s anisotropy on the ultrasonic testing results can be preliminarily assessed [32]. The experimental setup of the approach is shown in Figure 3.

#### 2.2.2. Description of CIVA Algorithms

The ultrasonic propagation simulation tools in the CIVA UT Simulation Tool are based on semi-analytical solutions used to calculate the ultrasonic field within a material and its interaction with flaws. CIVA utilizes ray tube theory [33] to model the acoustic field and the ray tracing method for visualization. The calculation of the field emitted by the transducer occurs by representing the field as a sum of fields created by elementary point sources on the surface of the piezoelectric plate. Using this method, it is possible to determine the time of flight, wave type transformations, and ray amplitudes. When calculating the amplitude and phase of the field reflected from defects, CIVA employs the Kirchhoff approximation method, variable separation method, and the geometric theory of diffraction [34].

#### 2.2.3. Model Description

To calculate the change in the signal amplitude from a defect for different directions of ultrasound propagation in an anisotropic material, an immersion ultrasonic model was employed. A flat-bottomed hole (FBH) with a 2 mm diameter located at a depth of 30 mm was used as a defect model for calculations. The general diagram of the model is presented in Figure 4. The amplitude of the reflected signal from the FBH was calculated for different positions of the material properties’ symmetry axes, simulating various directions of ultrasound propagation with respect to the properties’ symmetry axes. The calculation of the echo signal amplitude from the FBH was carried out by rotating the coordinate system (1, 2, 3) with a step of $15^{\circ }$ separately relative to each of the three axes, as shown in Figure 4a. A model of an unfocused piezoelectric transducer was used as a source of ultrasonic vibrations with a frequency of 5 MHz and a piezoelectric element diameter of 6 mm. The distance between the probe and the sample surface was 31 mm, ensuring that the near zone was completely in the immersion layer.

For each calculated value of the reflected FBH echo signal amplitude, the change in amplitude ΔA was determined relative to the amplitude value for the initial position of the axes shown in Figure 4. This calculation was performed for all studied materials using the measured single-crystal coefficients of the elasticity matrix as input data.

### 2.3. Assessment of Acoustic Anisotropy

#### 2.3.1. Methods Used to Assess Acoustic and Elastic Anisotropy

To conduct a comparative analysis of the effectiveness of different anisotropy coefficients for assessing amplitude changes obtained as a result of CIVA simulations, the following anisotropy coefficients were considered and calculated for the studied materials.

Many studies [35,36,37,38] have been devoted to assessing the anisotropy of the elastic properties of rocks. Similar to additive materials, rocks are generally orthotropic media; therefore, methods for assessing elastic anisotropy using acoustic methods can in some cases be applied to polycrystalline materials.

For transversely isotropic media, the following coefficients which allow for estimating the elastic anisotropy were proposed in [38]:(11)$$\delta =\frac{{\left(C_{13}+C_{44}\right)}^{2}-{\left(C_{33}-C_{44}\right)}^{2}}{2C_{33}\left(C_{33}-C_{44}\right)}\approx 4\left(\frac{\nu_{p}\left(\frac{\pi }{4}\right)}{\nu_{p}\left(0\right)}-1\right)-\left(\frac{\nu_{p}\left(\frac{\pi }{2}\right)}{\nu_{p}\left(0\right)}-1\right)$$
(12)$$\epsilon =\frac{C_{11}-C_{33}}{2C_{33}}\approx \frac{\nu_{p}\left(\frac{\pi }{2}\right)-\nu_{p}\left(0\right)}{\nu_{p}\left(0\right)}$$
(13)$$\gamma =\frac{C_{66}-C_{44}}{2C_{44}}\approx \frac{\nu_{SH}\left(\frac{\pi }{2}\right)-\nu_{SH}\left(0\right)}{\nu_{SH}\left(0\right)}$$
where $C_{ij}$ are the coefficients of the elasticity matrix; $\nu_{p}\left(0\right)$, $\nu_{p}\left(\frac{\pi }{4}\right)$, and $\nu_{p}\left(\frac{\pi }{2}\right)$ are the velocities of longitudinal waves, measured in the direction of the property’s symmetry axis at an angle of $45^{\circ }$ relative to the symmetry axis and perpendicular to the property’s symmetry axis, respectively; and $\nu_{SH}\left(0\right)$ and $\nu_{SH}\left(\frac{\pi }{2}\right)$ are the velocities of the transverse wave with horizontal polarization, measured in the direction of the property’s symmetry axis and perpendicular to its symmetry axis, respectively.

For orthotropic rock media, the following coefficients are used to estimate anisotropy based on the birefringence factor of shear waves [35]:(14)$$B_{1}=\frac{2(V_{1/2}-V_{1/3})}{V_{1/2}+V_{1/3}}=\frac{2(\sqrt{C_{55}}-\sqrt{C_{66}})}{\sqrt{C_{55}}+\sqrt{C_{66}}}$$
(15)$$B_{2}=\frac{2(V_{2/1}-V_{2/3})}{V_{2/1}+V_{2/3}}=\frac{2(\sqrt{C_{66}}-\sqrt{C_{44}})}{\sqrt{C_{66}}+\sqrt{C_{44}}}$$
(16)$$B_{3}=\frac{2(V_{3/1}-V_{3/2})}{V_{3/1}+V_{3/2}}=\frac{2(\sqrt{C_{55}}-\sqrt{C_{44}})}{\sqrt{C_{55}}+\sqrt{C_{44}}}$$
(17)$$B_{S}=\sqrt{B_{1}^{2}+B_{2}^{2}+B_{3}^{2}}$$
where $V_{i/j}$ is the velocity of the transverse wave in direction *i* with polarization *j*.

There is also a method for estimating the anisotropy of crystals [39] through the coefficient *G*:(18)$$G=\frac{2C_{44}}{C_{11}-C_{12}}={\left(\frac{\nu_{2}\left[100\right]}{\nu_{2}\left[110\right]}\right)}^{2}$$
where $\nu_{2}\left[100\right]$ and $\nu_{2}\left[110\right]$ are transverse wave velocities propagating along the [100] and [110] directions, respectively, with polarization perpendicular to [001].

#### 2.3.2. Proposed Method for Estimating Acoustic Anisotropy

In this work, we propose evaluating the anisotropy of additive samples using the birefringence factor, similar to the method described in [35], but focused on two directions of ultrasonic vibration propagation, i.e., $45^{\circ }$ and $90^{\circ }$ relative to the direction of the additive sample growth. Specifically, we introduce the following coefficients:$A_{13}$: The percentage ratio between the velocities of ultrasonic transverse waves with vertical (SV) and horizontal (SH) polarizations, propagating at a 45-degree angle to the growth direction.$A_{3}$: The similarity ratio for ultrasonic waves propagating at a 90-degree angle relative to the growth direction.$A_{g}$: The modulus of the difference between $A_{13}$ and $A_{3}$, indicating the extent of variation in the deviation between the velocities of transverse waves with mutually perpendicular polarizations in the two directions, i.e., at a 45-degree angle and normal to the growth direction.

These coefficients are determined with the following formulas:$$A_{13}=\frac{\left(\nu_{SH}\left(\frac{\pi }{4}\right)-\nu_{SV}\left(\frac{\pi }{4}\right)\right)}{\nu_{SH}\left(\frac{\pi }{4}\right)}\times 100\%=$$
(19)$$=\frac{\sqrt{C_{44}+C_{66}}-\sqrt{C_{44}+\frac{1}{2}\left(C_{33}+C_{11}\right)-\sqrt{\frac{1}{4}{\left(C_{11}-C_{33}\right)}^{2}+{\left(C_{13}+C_{44}\right)}^{2}}}}{\sqrt{C_{44}+C_{66}}}\times 100\%$$
(20)$$A_{3}=\frac{\left(\nu_{SH}\left(\frac{\pi }{2}\right)-\nu_{SV}\left(\frac{\pi }{2}\right)\right)}{\nu_{SH}\left(\frac{\pi }{2}\right)}\times 100\%=\frac{\sqrt{C_{44}}-\sqrt{C_{66}}}{\sqrt{C_{44}}}\times 100\%$$
(21)$$A_{g}=\left|A_{3}-A_{13}\right|$$
where $\nu_{SH}$ and $\nu_{SV}$ are the velocities of transverse waves with horizontal and vertical polarization, respectively. We propose using the coefficient $A_{g}$ as the general indicator of a material’s anisotropy. To calculate the coefficients, use a special cube-shaped sample with removed chamfers at $45^{\circ }$ relative to the growth direction, as shown in Figure 5.

## 3. Results

### 3.1. Ultrasonic Speed and Attenuation

The values of the velocities of longitudinal and transverse ultrasonic waves measured for various directions on the samples described in Section 2.1.1 are presented in Table 6. The values of the measured attenuation coefficients of longitudinal waves measured in three mutually perpendicular directions are presented in Table 7. The values of the calculated elasticity matrix coefficients $C_{ij}$ are provided in Table 8. It should be noted that in all the studied materials a difference of approximately 1–3% was noted between the measured velocities of longitudinal waves in the growth direction and normal to it.

The most significant velocity differences are noted for transverse waves. For instance, in EP648 SLM the velocity difference between waves $\nu_{1/3}$ and $\nu_{1/2}$ from Table 6 is about 17%. The largest deviation for longitudinal waves is observed in VG159 SLM, with a 10% difference between $\nu_{1/1}$ and $\nu_{13/13}$. The attenuation coefficients measured in three perpendicular directions (Table 7) are relatively low compared to heat-resistant alloys produced by traditional methods, where they are twice as high [29]. An F-test showed a statistically significant relationship between direction and attenuation coefficients for Inconel718 SLM, EP648 SLM, EP648 SLM+HIP, and VG159 SLM at a 5% significance level. This relationship can affect the amplitude of the reflected echo signal, which is important for ultrasonic testing of these materials.

### 3.2. Porosity Analysis

A small number of pores were found in almost every sample. The corresponding results are shown in Table 9.

In the Inconel 718 DLMD sample, a significant number of large-volume voids were observed; these were almost completely spherical in shape and were distributed evenly throughout the entire volume of the sample. The high level of porosity in Inconel 718 DLMD is likely the reason for the longitudinal wave velocities measured in three mutually perpendicular directions being lower than in the SLM sample of the same alloy. In addition, it should be noted that the attenuation coefficients of longitudinal waves in the Inconel 718 SLM sample are significantly lower than in the Inconel 718 DLMD sample. These results are consistent with studies of the relationship between the level of porosity of additive samples and the values of acoustic characteristics [4,5].

### 3.3. Maximum Change in Echo Signal Amplitude and Degree of Anisotropy

Using the method described in Section 2.2.3, changes in the amplitude ΔA of the echo signal from the FBH were calculated for the six samples and for varying values of the angle between the direction of propagation of vibrations and the properties’ axes of symmetry. Among the values of ΔA obtained by modeling, the values of the maximum changes in the amplitude $\Delta A_{\max}$ were selected for each material under study. To increase the set of data under study, we used the coefficients $C_{ij}$ from [13] for the Ti6Al4V alloy produced using SLM technology and metal forming (MF), as well as the elastic coefficients determined experimentally in [16] for the Inconel 718 and Inconel718NbC materials produced using DLMD technology. Calculations performed in CIVA showed that the largest changes in the amplitude of the echo signal occur when the angle between the propagation direction of the vibrations and the properties’ axis of symmetry co-directed with the direction of the material growth is approximately $45^{\circ }$ or $90^{\circ }$. For all samples, anisotropy coefficients and maximum changes in amplitude $\Delta A_{\max}$ were calculated using the methods described in Section 2.3. The values of the anisotropy coefficients and maximum changes in amplitude $\Delta A_{\max}$ are provided in Table 10 along with the maximum changes in amplitude and degree of anisotropy for each material.

To compare the effectiveness of various coefficients for estimating the value of $\Delta A_{\max}$, the correlation coefficients *R* between the degree of anisotropy and the change in amplitude were calculated. The values of the calculated correlation coefficients are provided in Table 11.

These values were calculated using the Pearson correlation coefficient *r* using the formula
(22)$$r=\frac{\sum \left(x_{i}-\bar{x}\right)\left(y_{i}-\bar{y}\right)}{n\sigma_{x}\sigma_{y}},$$
where $x_{i}$ and $y_{i}$ represent individual sample points, $\bar{x}$ and $\bar{y}$ are the means of the samples, $\sigma_{x}$ and $\sigma_{y}$ are the standard deviations of the samples, and *n* is the number of sample points.

This analysis allows for the determination of which anisotropy parameter most effectively predicts $\Delta A_{\max}$. By evaluating the Pearson correlation coefficient for each anisotropy parameter, it is possible to identify the parameter that exhibits the strongest linear relationship with the change in amplitude. As shown in Table 11, the results highlight the comparative effectiveness of these coefficients in estimating $\Delta A_{\max}$, providing insights for optimizing the evaluation of anisotropy in additive manufacturing materials.

It can be seen from Table 11 that the maximum value of 0.97 for the correlation coefficient was found for our proposed anisotropy coefficient $A_{g}$, which indicates that the coefficient $A_{g}$ has the strongest correlation with the change in amplitude.

### 3.4. Criteria for Classifying Anisotropic Additive Materials by Assessing the Magnitude of the Amplitude Change

To develop a criterion based on the $A_{g}$ coefficient which can allow for the classification of anisotropic materials depending on the magnitude of the change in amplitude, more data were needed. To solve this problem, data obtained as a result of numerically calculating the coefficients $C_{ij}$ from the system of Equations (13) and (14) for given values of the anisotropy coefficients $A_{13}$ and $A_{3}$, varying in the range from −15% to 15%, were added to the array of data in Table 7. The use of data obtained by modeling the parameters of anisotropic media was necessary in order to obtain estimated information about the relationship between the coefficient $A_{g}$ and the magnitude of the possible change in amplitude $\Delta A_{\max}$ in materials where the anisotropy coefficients $A_{13}$ and $A_{3}$ differ significantly from those calculated on experimental samples. The total set of data obtained as a result of modeling and experimental samples included 59 points. A scatterplot of the data between the anisotropy parameter $A_{g}$ and the magnitude of the maximum amplitude change $\Delta A_{\max}$ for the overall dataset is shown in Figure 6.

The scatterplot in Figure 6 shows that the data are distributed along a conditional inclined line (trend line), indicating a strong positive relationship between $\Delta A_{\max}$ and $A_{g}$. The regression equation has the following form:(23)$$\Delta A_{\max}=0.24A_{g}+0.48\;$$
while the $R^{2}$ coefficient equals 0.74. Significance testing for linear regression was performed using Fisher’s test of significance. At a 5% significance level, the *p*-value equals $2.4\times 10^{-13}<0.05$.

Based on Equation (Equation 23), the following criteria are proposed:If $A_{g}<6.5\%$, the estimated change in the amplitude of the echo signal reflected from the FBH with a 2 mm diameter and located at a depth of 30 mm does not exceed 2 dB when the propagation direction of the vibrations changes relative to the axis of the properties’ symmetry.If $6.5\%\leq A_{g}<23\%$, the estimated change in the amplitude of the echo signal reflected from the FBH with a 2 mm diameter and located at a depth of 30 mm can take values from 2 to 6 dB when the vibrations’ propagation direction changes with respect to the axis of the properties’ symmetry.If $A_{g}\geq 23\%$, the estimated change in the amplitude of the echo signal reflected from the FBH with a 2 mm diameter and located at a depth of 30 mm can take values in excess of 6 dB when the vibrations’ propagation direction changes relative to the axis of the properties’ symmetry. In this case, it is permissible to inspect the material by exciting ultrasonic waves only normally to the surface, with propagation along the axis of the material properties’ symmetry.

### 3.5. Deep Learning Approach for Predicting the Anisotropy Indicator $A_{g}$

Based on the data presented in Table 10, our objective in this section is to develop a machine learning model capable of accurately predicting the anisotropy indicator $A_{g}$. To achieve this, we employ an approach based on optimizing TensorFlow Keras neural network (NN) hyperparameters using a genetic algorithm (GA) [66,67].

A possible architecture of the suggested approach is shown in Figure 7. This architecture is designed to effectively capture the relationships between the input features and the target output. The input layer comprises nodes representing acoustic features. The architecture includes multiple hidden layers with varying numbers of neurons which are optimized via the GA. These hidden layers are responsible for learning complex patterns and interactions between the input features through nonlinear transformations. The output layer [68] contains a single node representing $A_{g}$. This output is the primary target of the model, providing insights into the effectiveness of ultrasonic testing under varying conditions. The architecture leverages the strengths of each input feature [69], transforming them through a series of hidden layers to generate an accurate prediction of the output. The specific design and depth of the hidden layers are determined through hyperparameter optimization, ensuring that the model captures the underlying relationships within the data.

The methodology involves defining the the hyperparameter space to consider various configurations, including different numbers of layers ranging from one to five, neurons per layer ranging from four to sixty-four in increments of four, and a diverse set of activation functions, including softmax [70], elu [71], selu [72], softplus [73], softsign [74], relu [75], tanh [76], sigmoid [77], hard sigmoid [78], exponential [79], and linear [80]. This defines the space in which the GA searches for the optimal hyperparameters.

The fitness function [81] used to build and trained the neural network is based on the following specified hyperparameters: layers, neurons per layer, and activation functions. The model is initialized with a specified input shape and trained using the Adam optimizer and mean squared error loss function for 100 epochs. The model’s performance can then be evaluated based on maximization of the R-squared (R^2^) score, calculated by comparing the true and predicted values on a test dataset comprising 20% of the overall sample.

The genetic algorithm (GA) optimizes the neural network’s hyperparameters through a series of structured operations, namely, population initialization, crossover, mutation, and parent selection.

The process begins by initializing a population of chromosomes, each of which represents a unique set of hyperparameters randomly selected from predefined search spaces. This initial population serves as the foundation for the evolutionary search process.

During the crossover phase, new chromosomes are generated by combining the hyperparameters of two parent chromosomes. This operation fosters genetic diversity within the population, as offspring inherit characteristics from both parents, potentially combining advantageous traits that enhance performance.

Mutation introduces variability into the population by randomly altering one or more hyperparameters within a chromosome [82]. This stochastic modification ensures a comprehensive exploration of the search space, preventing premature convergence on suboptimal solutions and enhancing the algorithm’s ability to discover globally optimal hyperparameters.

Parents are selected based on their fitness scores [83,84,85], which reflect the performance of the neural network using the corresponding set of hyperparameters. Several strategies exist for parent selection, such as roulette wheel selection, tournament selection, and rank-based selection, which ensure that higher-performing chromosomes have a greater probability of contributing to the next generation.

This iterative process of selection, crossover, and mutation is repeated over multiple generations. Each cycle refines the population to gradually optimize the hyperparameters and enhance the neural network’s predictive performance.

Through these iterative evolutionary operations, the genetic algorithm effectively navigates the hyperparameter space, converging on configurations that yield superior predictive accuracy and generalization capabilities for the neural network.

To optimize the parameters of the neural network for determining the anisotropy indicator, we ran several iterations of the GA with different parameters. For instance, Figure 8a shows the results of the first two generations, where the population size was 10, the number of generations was 3, and the number of parents for selection was 5. Figure 8b depicts the results of the third generation, where the parameters were altered; the population size remained 10, but the number of generations increased to 5 and the number of parents for selection was set to 10. Subsequently, Figure 8b displays the results for the fourth generation, where the population size was increased to 20, the number of generations to 10, and the number of parents for selection to 15. It can be observed that as the number of iterations increases, the $R^{2}$ score becomes more stable, indicating enhancement of the model’s performance.

The advantage of this method lies in its ability to automatically optimize the parameters of the neural network to achieve maximum prediction accuracy. This helps to reduce the time and resources spent on manual parameter tuning and ensures more stable and reliable results.

In order to validate the effectiveness of our method, we performed spatial cross-validation [86] on the neural networks with the best $R^{2}$ scores (>0.9).

Figure 9 depicts the $R^{2}$ scores obtained from spatial cross-validation, reflecting the performance of the neural network architecture. Each point in the plot corresponds to a specific combination of neurons and layers within the network. The plot captures the relationship between the number of neurons, the number of layers, and the resulting $R^{2}$ score, providing insights into the model’s predictive capability. Each data point represents the $R^{2}$ score obtained from spatial cross-validation for the respective neural network architecture, highlighting the influence of architectural choices on model performance.

Table 12 summarizes the key statistics of the $R^{2}$ scores obtained from different configurations. These configurations varied in terms of the number of layers, neurons per layer, and activation functions used. The median $R^{2}$ score, along with the standard deviation, minimum, and maximum $R^{2}$ values, provides insights into the overall predictive performance and variability across different network architectures.

## 4. Discussion

Traditional methods for evaluating the acoustic properties of materials often involve direct measurements of the velocity (*V*) and attenuation coefficient (α) of ultrasound waves propagating through the material [87,88]. These measurements are fundamental in calculating acoustic parameters such as elastic moduli and anisotropy coefficients. The velocity *V* and attenuation coefficient α are related to the material’s properties through equations such as
$$V=\frac{\omega }{k}\;\mathrm{and}\;\alpha =\frac{\omega }{2Q},$$
where ω is the angular frequency of the wave, *k* is the wave number, and *Q* is the quality factor.

Another common approach involves employing standard elasticity models tailored for anisotropic materials [89,90]. These models often utilize elasticity tensors or compliance matrices to describe the material’s response to stress and strain. The elastic constants can be derived from these models using equations such as
$$\sigma_{ij}=C_{ijkl}\epsilon_{kl},$$
where $\sigma_{ij}$ and $\epsilon_{kl}$ represent the stress and strain components, respectively, and $C_{ijkl}$ denotes the elasticity tensor components.

Some methods rely on theoretical models such as elasticity theory or wave propagation models. These theoretical frameworks provide insights into how ultrasound signals interact with the material [91,92], allowing for prediction of wave behaviors based on mathematical and physical models.

In contrast to traditional methods, the approach proposed in this study introduces the $A_{g}$ coefficient, which is based on evaluating the ratio of the velocities of transverse ultrasound waves with perpendicular polarizations at specific angles relative to the material’s growth direction [93]. The proposed coefficient $A_{g}$ demonstrates a strong linear correlation with changes in defect echo signal amplitudes. This method allows for more precise identification and classification of anisotropic properties in materials produced by additive manufacturing technologies.

To ensure the robustness and reliability of the proposed method using the $A_{g}$ coefficient for assessing acoustic anisotropy in additive manufacturing materials, it is crucial to thoroughly analyze potential error sources that could influence the experimental results. Minor deviations in measurement angles or velocity measurements can significantly impact the accuracy of ultrasound wave velocity and polarization angle measurements, thereby affecting the calculated $A_{g}$ coefficient. Techniques for improving measurement accuracy include precise angle alignment and calibration methods.

Variations in sample preparation [94], such as surface roughness [95], thickness uniformity [96], and material porosity [97], can introduce discrepancies in ultrasound wave propagation characteristics. These variations may lead to inconsistent $A_{g}$ values between different samples. Moreover, the assumption of material homogeneity [98] across the sample volume may not always hold true, especially in complex additive manufacturing structures. Variations in material composition and microstructure could influence ultrasound wave propagation and subsequently impact $A_{g}$ measurements. Advanced imaging techniques and microstructural analysis can aid in accurately assessing materials’ homogeneity.

Additionally, external environmental conditions [99] such as temperature and humidity fluctuations during measurements can alter the material’s properties and affect ultrasound wave behavior. These factors must be carefully controlled or accounted for in order to minimize their influence on $A_{g}$ calculations and ensure reliable results.

Understanding potential error sources is necessary in order to assess their influence on the accuracy and reliability of $A_{g}$ coefficients and subsequent analyses. These errors may introduce uncertainties into the correlation between $A_{g}$ and defect echo signal amplitudes, potentially affecting the predictive capability of the method.

For instance, inaccuracies in velocity measurements could lead to misinterpretation of anisotropy levels, while inconsistencies in sample preparation might obscure the true anisotropic effects [100]. By acknowledging and mitigating these potential error sources through rigorous experimental protocols and calibration procedures, the reliability of $A_{g}$ as an anisotropy indicator can be enhanced.

## 5. Conclusions

Our study investigated the acoustic properties of samples made from Inconel 718, EP648, AISI 321, and VG159 produced using additive technologies. Ultrasound wave velocities and attenuation coefficients were measured for six samples, and the elasticity coefficient matrices were obtained.

A review of existing methods for assessing acoustic anisotropy was carried out, on the basis of which we propose a new anisotropy indicator, the $A_{g}$ coefficient. The method for estimating anisotropy through the $A_{g}$ coefficient is based on determining the ratio between the velocities of transverse waves of mutually perpendicular polarization measured in two directions: at a 45-degree angle, and normal to the growth direction.

A comparison of methods for assessing acoustic anisotropy was carried out for the problem of establishing a relationship between anisotropy coefficients and the magnitude of the change in the echo signal amplitude from a defect when the direction of ultrasonic wave propagation changes relative to the axis of the properties’ symmetry. The results showed that the proposed $A_{g}$ anisotropy index had the strongest linear relationship with the change in amplitude. The value of the correlation coefficient was equal to 0.97.

We propose that threshold values of the $A_{g}$ anisotropy index can make it possible to distinguish between anisotropic additive materials depending on the magnitude of the estimated change in the echo signal amplitude from the defect when the inspection angle changes relative to the axis of the properties’ symmetry.

Additionally, we propose an approach for predicting $A_{g}$ using deep learning. Specifically, based on the optimization of neural network architecture hyperparameters using a genetic algorithm, it is possible to identify the architecture that can most accurately predict the anisotropy. This approach leverages the power of machine learning to efficiently analyze complex relationships between material properties and ultrasonic inspection outcomes, offering a promising avenue for enhanced defect detection in additive manufacturing processes.

## Figures and Tables

**Figure 1 sensors-24-04488-f001:**
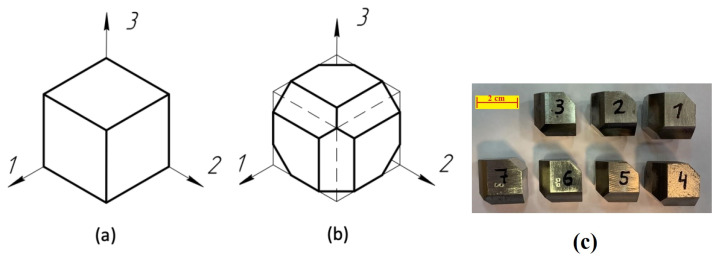
General view of the printed samples: (**a**) sample before machining; (**b**) sample after mechanical processing; (**c**) photos of sample after processing.

**Figure 2 sensors-24-04488-f002:**
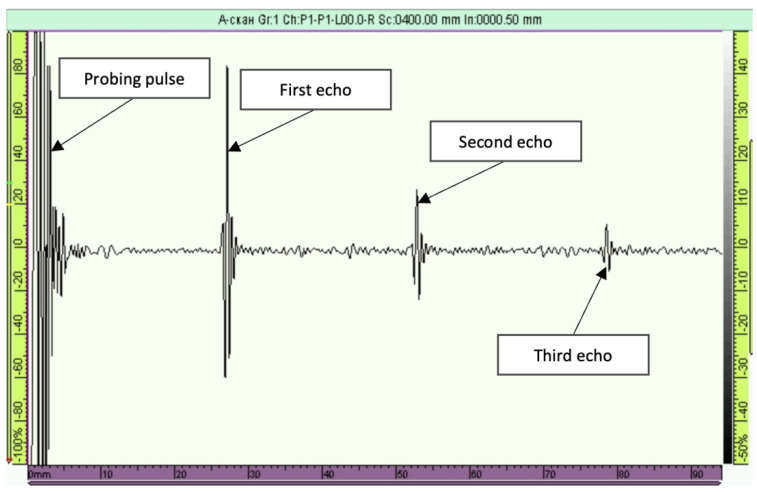
A-scan obtained for an Inconel 718 SLM sample at frequency f=5 MHz with the C543SM longitudinal wave transducer.

**Figure 3 sensors-24-04488-f003:**
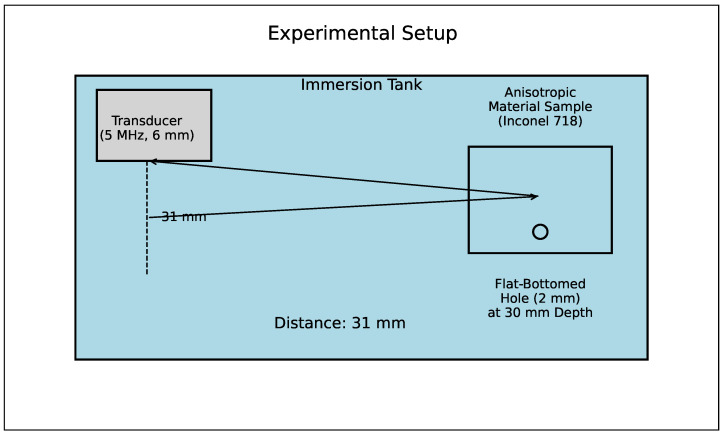
Schematic diagram of the experimental setup for assessing changes in echo signal amplitude.

**Figure 4 sensors-24-04488-f004:**
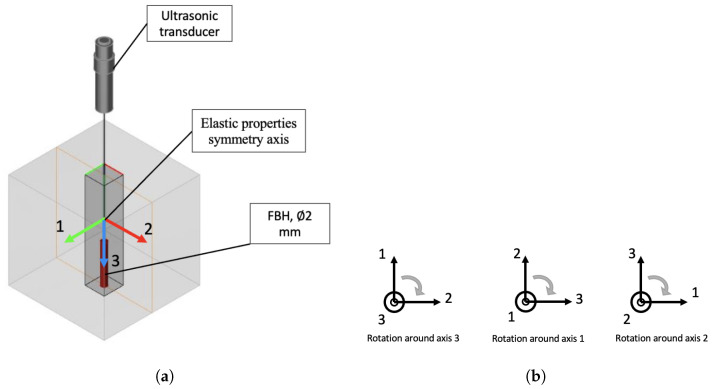
(**a**) Model diagram for calculation of amplitude changes and (**b**) rotation of property symmetry axes.

**Figure 5 sensors-24-04488-f005:**
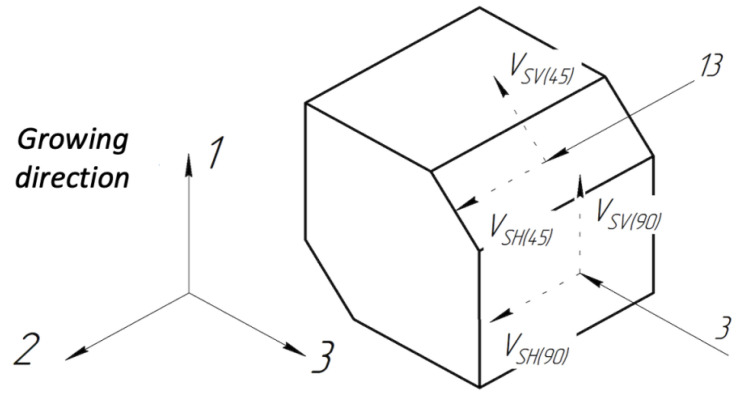
Sample for determining anisotropy coefficients using Formulas (Equation 19)–(Equation 21).

**Figure 6 sensors-24-04488-f006:**
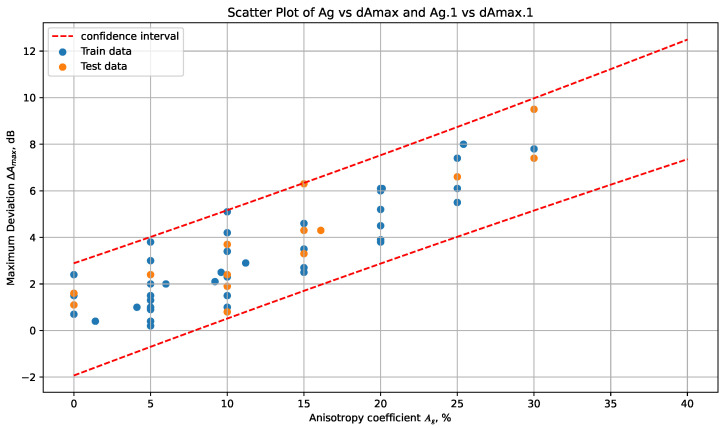
Scatterplot of the data between the anisotropy parameter $A_{g}$ and the magnitude of the maximum amplitude change $\Delta A_{\max}$.

**Figure 7 sensors-24-04488-f007:**
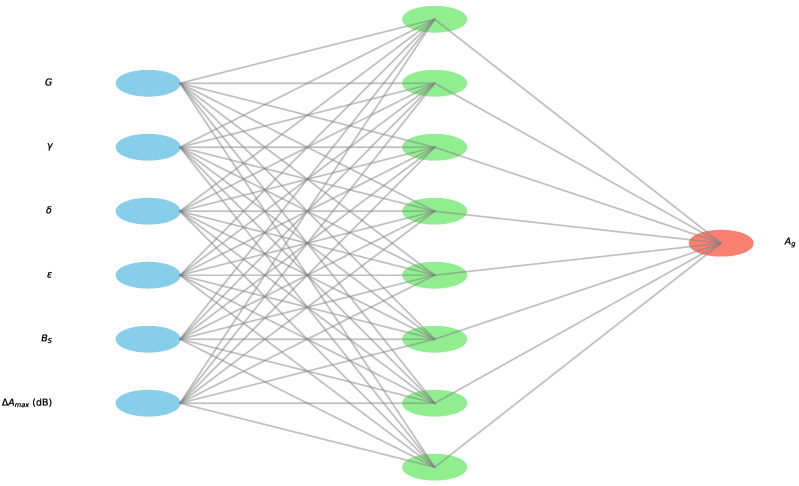
Example showing possible optimized NN architecture.

**Figure 8 sensors-24-04488-f008:**
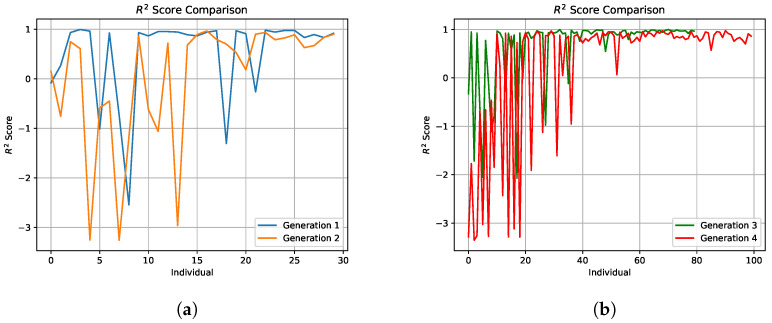
Change in $R^{2}$ score as a function of the number of GA generations during fine-tuning of the neural network hyperparameters: (**a**) the orange and blue curves represent experiments with a population size of 10, three generations, and five parents selected per generation; (**b**) the red curve corresponds to experiments with a population size of 20, ten generations, and fifteen parents selected per generation, while the green curve indicates experiments with a population size of 10, eight generations, and ten parents selected per generation.

**Figure 9 sensors-24-04488-f009:**
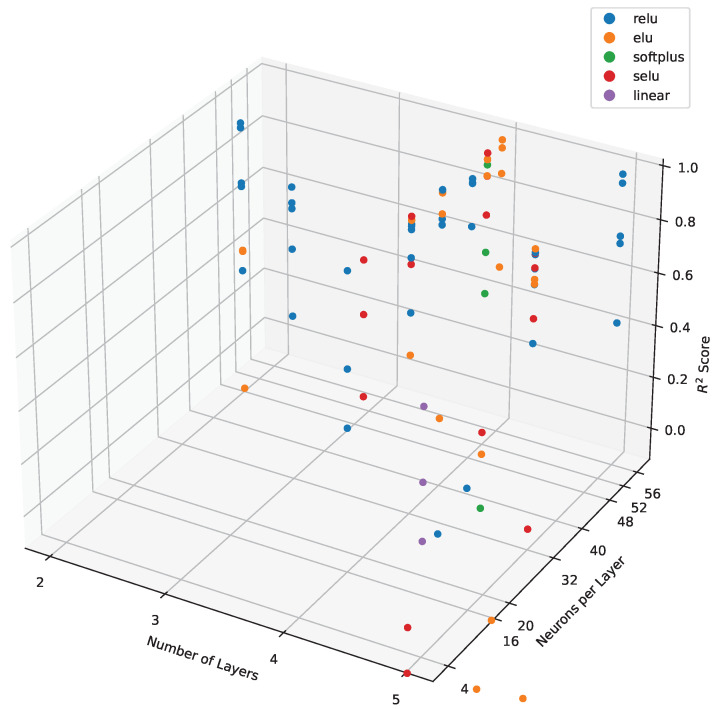
Cumulative $R^{2}$ scores obtained from spatial cross-validation for various neural network architectures, represented by different combinations of neurons and layers.

**Table 1 sensors-24-04488-t001:** Summary and limitations of approaches to studying anisotropy and quality control in additive manufacturing.

Source	Proposed Approach	Limitations
Sridharan et al. [10]	Investigation of ultrasonic atomization of metals using high-speed imaging to control particle size and shape.	Research is limited to aluminum, and the influence of cavitation events needs further study for other metals.
Du et al. [11]	Use of ultrasonic methods to measure anisotropy in nickel alloy IN718 fabricated by AM and DLD.	Methodology focuses on a specific material (IN718) and needs validation for other materials and additive manufacturing methods.
Sol et al. [12]	Application of pulse-echo ultrasonic testing to study anisotropy in $AlSi_{10}Mg$ samples produced by selective laser melting (SLM).	Limited to one type of material ($AlSi_{10}Mg$) and requires additional studies for other additive materials.
Thevet et al. [13]	Study of anisotropic elastic properties of $Ti_{6}Al_{4}V$ alloy using dynamic pulse-echo ultrasonic techniques.	Focuses on $Ti_{6}Al_{4}V$ alloy and needs validation for other materials and manufacturing methods.
Gou et al. [14]	Use of ultrasonic peening treatment to enhance anisotropy of mechanical properties and microstructure in cold metal transfer process of $Ti_{6}Al_{4}V$.	Primarily addresses one process (cold metal transfer) and material ($Ti_{6}Al_{4}V$), requiring further research for other processes and materials.
Sun et al. [15]	Inter-layer ultrasonic impact strengthening in wire and arc additive manufacturing (WAAM) to reduce anisotropy.	Limited to one manufacturing process (WAAM) and needs validation on other additive manufacturing methods and materials.
Markanday et al. [16], Aleshin et al. [6,17], Li et al. [18], Kransutskaya et al. [2]	Investigation of anisotropic material properties dependency on direction using ultrasonic methods.	Combines different studies and materials but requires a systematic approach for standardizing methods and generalizing results.
Simonelli et al. [20], Sufiiarov et al. [21]	Study of crystallographic texture and its impact on anisotropy of materials produced by additive manufacturing methods.	Textural studies focus on a few materials and methods, requiring broader validation.

**Table 2 sensors-24-04488-t002:** Growth parameters for SLM samples.

Sample No	Growth Technology	Material	Laser Parameters		
			Power, V	Scan Speed, mm/s	Hatch Type
1	SLM	EP648	170	800	Solid
2	SLM + HIP	EP648	180	800	Staggered
3	SLM	Inconel 718	180	700	Staggered
4	SLM	VG159	300	805	Staggered

**Table 3 sensors-24-04488-t003:** Growth parameters for the DLMD samples.

Sample No	Growth Technology	Material	Power, V	Process Speed, mm/c	Spot Diameter, mm	Layer Step, mm
5	DLMD	AISI 321	2200	25	3.2	0.6
6	DLMD	Inconel 718	1570	25	2.2	0.6

**Table 4 sensors-24-04488-t004:** Density values used to calculate $C_{ij}$ coefficients.

Material	ρ, g/cm^3^
EP648	8.0
Inconel 718	8.2
VG159	8.2
AISI 321	7.9

**Table 5 sensors-24-04488-t005:** X-ray CT performance modes.

Sample Number	Cathode Voltage, kV	Current, μA	Focal Spot Size, μm	Number of Projections	Voxel Size, μm
VG159 SLM	220	70	15.4	1700	15.38
AISI321 DLMD	220	70	15.4	1700	15.22
EP648 SLM	220	70	15.4	1700	15.45
EP648 SLM+HIP	220	70	15.4	1700	15.22
Incinel718 DLMD	220	70	15.4	1700	15.22
Inconel718 SLM	220	70	15.4	1700	15.22

**Table 6 sensors-24-04488-t006:** Ultrasound wave velocity values.

Measured Velocity, km/s	EP648 SLM	EP648 SLM+HIP	Inconel718 DLMD	Inconel718 SLM	VG159 SLM	AISI 321 DLMD
$\nu_{\left(1/1\right)}$	5.69	5.82	5.40	5.62	5.53	5.66
$\nu_{\left(2/2\right)}$	5.80	5.95	5.60	5.72	5.48	5.75
$\nu_{\left(3/3\right)}$	5.80	5.92	5.57	5.72	5.48	5.73
$\nu_{\left(3/2\right)}$	2.64	2.84	2.92	2.96	3.42	2.96
$\nu_{\left(1/3\right)}$	3.08	3.14	3.10	3.25	3.48	3.18
$\nu_{\left(1/2\right)}$	3.07	3.17	3.04	3.25	3.44	3.20
$\nu_{\left(13/13\right)}$	5.78	5.89	5.60	5.85	6.03	5.77
$\nu_{\left(12/12\right)}$	5.82	5.92	5.56	5.85	6.03	5.79
$\nu_{\left(23/23\right)}$	5.54	5.76	5.57	5.72	5.99	5.62

**Table 7 sensors-24-04488-t007:** Values of the longitudinal wave attenuation coefficients f=5 MHz, dB/cm.

Material	EP648 SLM	EP648 SLM+HIP	Inconel718 DLMD	Inconel718 SLM	VG159 SLM	AISI 321 DLMD
Direction						
1	0.17	0.5	1.66	0.11	0.63	0.72
2	0.12	0.3	1.59	0.32	0.83	0.59
3	0.43	0.14	1.61	0.25	1.18	0.69

**Table 8 sensors-24-04488-t008:** Values of the elasticity matrix coefficients $C_{ij}$, GPa.

Coefficients $C_{\mathrm{ij}}$	EP648 SLM	EP648 SLM+HIP	Inconel718 DLMD	Inconel718 SLM	VG159 SLM	AISI 321 DLMD
$C_{11}$	259	271	259	257	251	253
$C_{22}$	269	283	269	255	246	261
$C_{33}$	269	281	269	239	246	259
$C_{44}$	56	66	72	76	96	71
$C_{55}$	75	79	87	75	98	79
$C_{66}$	75	79	87	70	97	81
$C_{23}$	110	119	125	115	151	97
$C_{13}$	121	121	125	108	152	113
$C_{12}$	127	125	125	113	154	110

**Table 9 sensors-24-04488-t009:** Values of porosity analysis.

Sample	Volume Porosity (%)	Presence of Cracks	Other Defects	Maximum Defect Volume (mm^3^)	Quantity of Founded Defects
VG59 SLM	0.00087	No	No	0.000419	47
AISI 321 DLMD	0.00061	No	No	0.001291	25
EP648 SLM	0.00113	No	No	0.000872	78
EP648 SLM+HIP	0	No	No	-	-
INCONEL718 DLMD	4.97	No	No	0.03789	51,638
Inconel 718 SLM	0.00002	No	No	0.000158	2

**Table 10 sensors-24-04488-t010:** Maximum changes in amplitude and degree of anisotropy.

Material	$\Delta A_{\max}$ (dB)	$B_{S}$	ϵ	δ	γ	*G*	$A_{g}$ (%)
EP648 SLM	2.9	0.21	0.02	0.071	−0.126	0.94	11.2
EP648 SLM+HIP	2	0.14	−0.004	−0.023	−0.001	0.99	6
Inconel718 SLM	4.3	0.14	0.018	0.168	−0.084	1.2	16.1
Inconel718 DLMD	2.1	0.06	0.036	0.097	−0.043	1.06	9.2
VG159 SLM	9.5	0.02	−0.01	0.518	−0.01	2.06	30
AISI 321 DLMD	2.5	0.09	−0.011	−0.035	0.085	0.97	9.6
Ti6Al4V SLM [13]	1	0.1	0.026	−0.019	0.075	0.86	4.11
Ti6Al4V MF [13]	0.4	0	0	−0.019	0	1.02	1.4
Inconel 718 DLMD [16]	6.1	0.21	0.057	0.066	−0.131	0.9	20.1
Inconel718NbC DLMD [16]	8	0.26	0.074	0.058	−0.159	0.88	25.4
AlSi10Mg SLM [40,41,42]	3.4	0.19	0.012	0.049	−0.097	1.1	12.5
316L SLM [43,44,45]	2.2	0.15	0.008	0.35	−0.065	1.03	9.0
17-4 PH SLM [46,47]	2.7	0.13	0.02	0.067	−0.089	0.96	10.4
CoCrW SLM [48,49]	3.9	0.18	0.015	0.056	−0.105	1.08	14.3
Hastelloy X SLM [50,51,52]	4.6	0.22	0.023	0.074	−0.111	1.2	17.0
18Ni-300 MS [53,54,55]	3.1	0.17	0.019	0.061	−0.092	1.05	13.2
AlSi12 SLM [41,56]	2.8	0.16	0.011	0.048	−0.086	1.09	11.7
CuSn10 SLM [57,58,59]	3.5	0.20	0.014	0.055	−0.095	1.12	13.6
Al7075 SLM [60,61]	4.0	0.21	0.018	0.062	−0.108	1.15	15.2
Al2024 SLM [62,63]	3.6	0.19	0.016	0.057	−0.102	1.11	13.8
NiTi SLM [64,65]	4.4	0.23	0.025	0.073	−0.114	1.17	16.8

**Table 11 sensors-24-04488-t011:** Correlation coefficients between the degree of anisotropy and change in amplitude.

	ϵ	δ	γ	*G*	$B_{S}$	$A_{g}$
$\Delta A_{\max}$	0.31	0.74	−0.50	0.58	0.31	0.97

**Table 12 sensors-24-04488-t012:** Summary statistics of $R^{2}$ scores for different neural network configurations.

Number of Layers	Neurons per Layer	Activation Function	Median $R^{2}$	Standard Deviation of $R^{2}$	Minimum $R^{2}$	Maximum $R^{2}$
2	48	relu	0.699986	0.205056	0.360193	0.927981
5	32	relu	0.928194	0.120986	0.657188	0.986920
4	32	relu	0.949115	0.123375	0.643705	0.968744
3	32	relu	0.906979	0.172669	0.506309	0.985874
5	56	relu	0.757946	0.199017	0.430448	0.987120

## Data Availability

The original contributions presented in the study are included in the article, further inquiries can be directed to the corresponding authors.

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
