# Peer review of "Assessment of Anisotropic Acoustic Properties in Additively Manufactured Materials: Experimental, Computational, and Deep Learning Approaches"

_sensors, 2024, doi:10.3390/s24144488_

Round 1
Reviewer 1 Report
Comments and Suggestions for Authors
The manuscript "Assessment of Anisotropic Acoustic Properties in Additively Manufactured Materials: Experimental, Computational, and Deep Learning Approaches" investigates the effects of anisotropic elasticity constant on defect signal amplitudes in additive technologies products. Six samples are made and wave velocities are measured to obtain the elasticity coefficient. Overall, this work is interesting and the content is well-organized. It integrates theoretical foundations and experimental investigations. I think that readers will find this article particularly interesting and valuable, although the presentation of work results is not intuitive and detailed enough, which is suggested to be improved in future work. I recommend the publication of this work.
Author Response
Dear Reviewer,
We would like to express our sincere gratitude for taking the time to review our manuscript "Assessment of Anisotropic Acoustic Properties in Additively Manufactured Materials: Experimental, Computational, and Deep Learning Approaches". Your feedback and suggestions are invaluable to us, and we appreciate your expertise and insights.We are pleased to hear that you found our work interesting and well-organized, and that you appreciate the integration of theoretical foundations and experimental investigations.
.Sincerely, Ivan.
Reviewer 2 Report
Comments and Suggestions for Authors
It is a research paper instead of a high school project report.
There are many English mistakes in the paper. Authors must take it serious.
After rewriting the paper, please resubmit it again.
Comments on the Quality of English LanguageYou emphasized Ag in the paper. It will be better that you list the variable at the first.
Before:
Parameters A13, A3, and Ag, derived from wave velocity relationships, characterized 6 acoustic anisotropy.
After:
Parameters Ag, A3, and A13, derived from wave velocity relationships, characterized 6 acoustic anisotropy.
Another examples are missing key information as follows:
1) At Lines 181-182, authors says: "The measurement technique 181 corresponded to that described in [? ], method 2."
2) At lines 257, authors says:"...axes shown in Figure ??. This calculation was performed for all studied materials using the"
3) at Line 279, author says:"There is also a known method for estimating the anisotropy of crystals [? ] through"
Author Response
Dear Reviewer2,
I am writing to express my sincere gratitude for your thorough and insightful review of my article. Your detailed comments and constructive suggestions have been invaluable in improving the quality and clarity of the manuscript.
Attached, please find a file with point-by-point responses to each of your comments. I hope these revisions adequately address your concerns and enhance the overall quality of the paper.
Thank you for your time and effort in providing such thoughtful feedback.
Best regards, Ivan.

Reviewer 3 Report
Comments and Suggestions for Authors
In this work, the authors explored the effect of anisotropy on the reliability of ultrasonic testing, especially in additive manufacturing materials. The method of measuring elastic constants and the method of using CIVA software for computational simulation are described in detail. Through a large number of experiments and computational simulations, the influence of anisotropy on the amplitude of ultrasonic signals is analyzed in detail, and a method of predicting anisotropy coefficients by deep learning is proposed, which provides an innovative solution to reduce the computational cost. The comments are as follows :
1. The writing logic of the article is not very smooth. It is recommended to adjust the logical structure. The method part can be described in a concise language.
2. When describing the method of measuring elastic constants and anisotropic coefficients, the schematic diagram of experimental settings can be added to help readers understand the experimental process more intuitively.
3. It is suggested to increase the analysis of the potential error sources in the experiment and discuss its possible influence on the results.
4. It is recommended to increase the statistical analysis of the data, such as the use of standard deviation, confidence interval, etc., to enhance the reliability and persuasiveness of the data.
5. It is recommended to increase the description of the architecture, training data, and training process of the deep learning model.
6. Check in detail the expression of the references, figure numbers, and table numbers in the article, such as lines 182, 191, 213, and 246. And unify the table format.
Author Response
Dear Reviewer3,
I am writing to express my sincere gratitude for your thorough and insightful review of our article. Your detailed comments and constructive suggestions have been invaluable in improving the quality and clarity of the manuscript.
Thank you for your time and effort in providing such thoughtful feedback.
Please find attached file with point-by-point answers on your suggestions.
Best regards,Ivan.

Reviewer 4 Report
Comments and Suggestions for Authors
The study addresses the impact of acoustic anisotropy on the reliability of ultrasonic testing for additive manufacturing (AM) products, focusing on how elasticity constants of anisotropic materials influence defect signal amplitudes. I think this article is well-written. Here are my suggested revisions:
- Introduction: Please further explain the scientific basis for using deep learning in this section.
- Figure 1(c): Please add a scale bar.
- Line 182: The ref is missing, please add it.
- Line 343: How were the calculated correlation coefficients determined, and what is the basis for their calculation?
- Someformulas in the manuscript are not numbered, please confirm if this is acceptable. For example, on line 368, the reviewer could not find where equation 16 is.
- Conclusions: In this section, could you compare this model with traditional models and clarify its advantages and scientific significance?
Author Response
Dear Reviewer4,
I am writing to express my sincere gratitude for your thorough and insightful review of my article. Your detailed comments and constructive suggestions have been invaluable in improving the quality and clarity of the manuscript.
Attached, you will find a document with point-by-point responses to each of your comments. I hope this addresses all your concerns and demonstrates how your feedback has been incorporated into the revised manuscript.
Thank you for your time and effort in providing such thoughtful feedback.
Best regards, Ivan.

Round 2
Reviewer 2 Report
Comments and Suggestions for Authors
It has been improved. Great job.
Reviewer 3 Report
Comments and Suggestions for Authors
The comments are responded correctly.